# Lacrimal Gland Epithelial Cells Shape Immune Responses through the Modulation of Inflammasomes and Lipid Metabolism

**DOI:** 10.3390/ijms24054309

**Published:** 2023-02-21

**Authors:** Vanessa Delcroix, Olivier Mauduit, Menglu Yang, Amrita Srivastava, Takeshi Umazume, Cintia S. de Paiva, Valery I. Shestopalov, Darlene A. Dartt, Helen P. Makarenkova

**Affiliations:** 1Department of Molecular Medicine, The Scripps Research Institute, 10550 North Torrey Pines Road, La Jolla, CA 92037, USA; 2Schepens Eye Research Institute/Massachusetts Eye and Ear, Department of Ophthalmology, Harvard Medical School, Boston, MA 02114, USA; 3The Ocular Surface Center, Cullen Eye Institute, Department of Ophthalmology, Baylor College of Medicine, Houston, TX 77030, USA; 4Bascom Palmer Eye Institute, University of Miami School of Medicine, Miami, FL 33136, USA

**Keywords:** lacrimal gland, inflammasome, lipid metabolism, Sjogren’s syndrome, dry eye, regeneration, acute inflammation, chronic inflammation

## Abstract

Lacrimal gland inflammation triggers dry eye disease through impaired tear secretion by the epithelium. As aberrant inflammasome activation occurs in autoimmune disorders including Sjögren’s syndrome, we analyzed the inflammasome pathway during acute and chronic inflammation and investigated its potential regulators. Bacterial infection was mimicked by the intraglandular injection of lipopolysaccharide (LPS) and nigericin, known to activate the NLRP3 inflammasome. Acute injury of the lacrimal gland was induced by interleukin (IL)-1α injection. Chronic inflammation was studied using two Sjögren’s syndrome models: diseased *NOD.H2^b^* compared to healthy *BALBc* mice and Thrombospondin-1-null (TSP-1^-/-^) compared to TSP-1^WT^ *C57BL/6J* mice. Inflammasome activation was investigated by immunostaining using the *R26^ASC-citrine^* reporter mouse, by Western blotting, and by RNAseq. LPS/Nigericin, IL-1α and chronic inflammation induced inflammasomes in lacrimal gland epithelial cells. Acute and chronic inflammation of the lacrimal gland upregulated multiple inflammasome sensors, caspases 1/4, and interleukins *Il1b* and *Il18*. We also found increased IL-1β maturation in Sjögren’s syndrome models compared with healthy control lacrimal glands. Using RNA-seq data of regenerating lacrimal glands, we found that lipogenic genes were upregulated during the resolution of inflammation following acute injury. In chronically inflamed *NOD.H2^b^* lacrimal glands, an altered lipid metabolism was associated with disease progression: genes for cholesterol metabolism were upregulated, while genes involved in mitochondrial metabolism and fatty acid synthesis were downregulated, including peroxisome proliferator-activated receptor alpha (PPARα)/sterol regulatory element-binding 1 (SREBP-1)-dependent signaling. We conclude that epithelial cells can promote immune responses by forming inflammasomes, and that sustained inflammasome activation, together with an altered lipid metabolism, are key players of Sjögren’s syndrome-like pathogenesis in the *NOD.H2^b^* mouse lacrimal gland by promoting epithelial dysfunction and inflammation.

## 1. Introduction

Dry eye disease affects millions of adults worldwide and can be divided into two major types: evaporative dry eye and aqueous-deficient dry eye (ADDE) [1,2]. ADDE is characterized by a reduced secretion or an altered composition of fluid from the lacrimal gland (LG) [3,4], the exocrine tubuloacinar gland responsible for secreting the aqueous layer of the tear film [5,6,7]. ADDE induces eye irritation and pain, and may lead to severe ocular surface disorders [8,9]. The leading cause of ADDE is the chronic inflammation of the LG triggered by aging or auto-immune diseases such as Sjögren’s syndrome (SS).

Dry eye disease involves inflammatory mechanisms and the production of several tear cytokines, including interleukins (IL)-1α and IL-1β, which correlate with clinical severity [10]. IL-1α/β are potent proinflammatory cytokines that function as key danger signals during infection or tissue damage. IL-1α/β binding to their receptor IL-1R1 promotes the transcription of genes involved in acute and chronic inflammation. In mice, reversible ADDE can be experimentally induced by a single injection of IL-1α into the LG [11,12]. Several studies demonstrated that IL-1α induces acute LG inflammation and destruction within the first two days after the injury [11,12,13,14]. The resolution of inflammation was noted on the third day after injury and was followed by cell proliferation and complete regeneration within 5–7 days [12,13]. 

Inflammasomes are large intracellular multiprotein complexes that play a central role in innate immunity [15,16]. The largest class of inflammasomes contain an apoptosis-associated speck-like protein containing a CARD (ASC, encoded by the gene *Pycard*) and pathogen/danger sensors, which recruit and activate the pro-caspase 1 (pro-CASP1). Each type of inflammasome is characterized by a particular sensor or receptor: PYRIN, the nucleotide-binding domain (NOD), leucine-rich repeat (LRR)-containing protein (NLR) family (e.g., NLRP1/3/6, NLRC4), or the pyrin and HIN domain-containing protein (PYHIN) family (e.g., AIM2 and IFI204, the murine homolog of human IFI16). Canonical inflammasomes cleave IL-1β and IL-18 precursors to generate the mature cytokines. Activated CASP1 and the non-canonical inflammasome formed by CASP4/11 can also cleave the pore-forming protein gasdermin D (GSDMD), which mediates interleukin secretion [17]. GSDMD is required for pyroptosis, an immunogenic form of cell death which enables the massive release of active IL-1α and IL-1β from dying cells [15,18]. Inflammasomes can be activated by a multitude of infectious and sterile stimuli, including microbiome-derived signals and host-derived signals, and were found in different cell types [19]. We previously showed that acute LG inflammation triggers the upregulation of inflammasome-related molecules: *Casp4*, *Nlrp3*, the purinergic receptors P2RX7 and P2RY2, the Pannexin-1 (*Panx1*) membrane channel glycoprotein—a key regulator of inflammasome assembling—, and numerous proinflammatory factors including IL-1β and IL-18 [14]. 

Whilst inflammasome signaling is critical for the initiation of a fast innate immune response to tissue damage or invading pathogens, aberrant inflammasome activation contributes to various pathologies, including autoimmune disorders, cardiometabolic diseases, cancer, and neurodegenerative diseases [20,21]. It has been shown that NLRP1, NLRP3, NLRC4, and AIM2 inflammasomes play a significant role in shaping immune responses and regulating the homeostasis of intestine and ocular surface immunity in several inflammatory diseases [22,23,24,25,26,27]. In human patients suffering from SS dry eye, conjunctival impression cytology demonstrated an upregulation of *Nlrp3* and *Casp1* [25]. Baldini and coauthors [28] showed that in the salivary glands of SS patients, the increased expression of NLRP3, CASP1, and P2RX7 was a marker of disease and correlated with the focus score evaluating the number of immune cell infiltrates in gland sections. Moreover, AIM2 inflammasomes are activated in the salivary epithelium of primary Sjögren’s syndrome (pSS) patients and correlate with the expression levels of type I interferon (IFN) signature genes [29]. IFNs are major regulators of the innate immune response and contribute to the activation of canonical and non-canonical inflammasomes [30], partly by inducing the guanylate-binding proteins (GBPs) that are dynamin-like GTPases [31,32]. Among them, GBP1 and GBP2 are upregulated in the biopsies of pSS salivary glands [33] and the latter was proposed as a biomarker for SS in saliva [34] and salivary glands [35]. Although the implication of inflammasome modulators in chronically inflamed LG remains unknown, their therapeutic potential is supported by our observation that the inhibition of *Panx1* or *Casp4* reduced the inflammation of LGs from thrombospondin-1-null *(TSP-1*^-/-^) mice, a model for SS, and improved the epithelium repair through the increased engraftment of progenitor cells [14].

In this study, we investigated inflammasome formation during acute and chronic LG inflammation using the *R26^ASC-citrine^* mouse. This reporter mouse forms fluorescent ASC specks when inflammasomes are activated [36]. We show that LG epithelial cells can form inflammasomes upon sensing danger signals and inflammation. To identify inflammasome types involved in acute and chronic inflammation, we analyzed RNA-sequencing (RNA-seq) data from a previously published study on acute injury [13] and performed RNA-seq of LGs from *NOD.H2^b^* mice [37]—a pSS mouse model. Our results indicate a strong activation of multiple inflammasome complexes during acute and chronic inflammation. Finally, we analyzed the RNA-seq data of the LGs during acute and chronic inflammation in terms of the biological pathways to identify candidate mechanisms that promote the resolution of inflammation. We demonstrated that lipid biosynthesis is activated during the resolution of inflammation/regeneration after acute injury, but that genes for fatty acid and cholesterol synthesis are, respectively, down- and upregulated during chronic inflammation. Moreover, chronic inflammation also downregulates the genes involved in the tricarboxylic acid (TCA) cycle and the β-oxidation of fatty acids in the mitochondria. Combined, our results show that during chronic inflammatory disease, LG epithelial cells have reduced lipid biosynthesis, accumulated cholesterol, and showed mitochondrial dysfunction. Altogether, these alterations likely induce cell damage, sustain inflammasome activation, and impair LG regeneration and function. 

## 2. Methods

### 2.1. Mice

For the in vivo detection of the activated inflammasome complex, we employed the transgenic mouse expressing a mouse ASC-citrine fusion protein in the Rosa26 (R26) locus (*R26^ASC-citrine^*, kind gift of Dr. Golenbock) [36]. *R26^ASC-citrine^* mice were bred and maintained on the *C57BL/6J* background. 

*B6.129S2-Thbs1^tm1Hyn^/J* mice (*TSP-1*^-/-^, RRID:IMSR_JAX:006141) were originally purchased from Jackson Laboratory (Sacramento, CA, USA) and were bred and maintained on the *C57BL/6J* background. For immunoblotting, *TSP-1*^-/-^ mice were compared to age-matched wild-type (WT) *C57BL/6J* mice. For immunofluorescence, we crossed *R26^ASC-citrine^* x *TSP-1*^-/-^ mice to obtain *TSP-1*^-/-^:*R26^ASC-citrine^* mice that were compared to age-matched *R26^ASC-citrine^* mice. The *NOD.B10Sn-H2^b^/J* mice (*NOD.H2^b^*, RRID:IMSR_JAX:002591) and their *BALB/cJ* controls (*BALBc*, RRID:IMSR_JAX:000651) mice were purchased from Jackson Laboratory (Sacramento, CA, USA).

The mice were housed under standard conditions of temperature and humidity, with a 12 h light/dark cycle and free access to food and water. All experiments were performed in compliance with the ARVO Statement for the Use of Animals in Ophthalmic and Vision Research and the Guidelines for the Care and Use of Laboratory Animals, published by the US and National Institutes of Health (NIH Publication No. 85-23, revised 1996), and were pre-approved by TSRI Animal Care and Use Committee.

### 2.2. Induction of Inflammasome Formation with Lipopolysaccharide and Nigericin

*R26^ASC-citrine^* mice were primed by intraglandular injection of lipopolysaccharide (LPS, 1 µg/mL). After 3 h, inflammasome activation was induced by the injection of nigericin (10 µM). Six hours after the last injection, the mice were sacrificed and LGs were dissected out and processed for frozen section preparation.

### 2.3. LG Acute Injury 

For acute injury experiments, the *R26^ASC-citrine^* and WT *C57BL/6J* females were used. LG inflammation in these mice was induced by the intraglandular injection of IL-1α, as previously described [11]. Briefly, 12 female mice (10 to 12 weeks old) were anesthetized, and the exorbital LG was injected with either saline (vehicle) or IL-1α (1 μg; PeproTech, Cranbury, NJ, USA) in a total volume of 2 μL using a Hamilton glass syringe (#300329, World Precision Instruments, Inc., Sarasota, FL, USA) and NanoFil 35G needle (#NF35BV-2, World Precision Instruments, Inc., Sarasota, FL, USA). The LGs from uninjected mice were used as additional controls. The LGs were harvested 6 and 12 h after injection and processed for immunohistochemistry and RNA extraction. 

### 2.4. Frozen Section Preparation and Immunostaining

The dissected LGs were fixed with 2% paraformaldehyde in PBS (pH 7.4) for 45 min and frozen in 2-methylbutane cooled by liquid nitrogen, and 10-μm cryosections were prepared using Hacker/Bright OTF5000-LS004 Cryostat. The sections were blocked with 1% bovine serum albumin in Tris-buffered saline containing 0.05% Tween 20. The following primary antibodies were used for immunostaining overnight at 4 °C: mouse monoclonal α-smooth muscle actin antibody (1/200, clone 1A4; #A2547, RRID:AB_476701, Millipore-Sigma, Rocksville, MD, USA) was used to label the myoepithelial cells (MECs) and pericytes (contractile cells around the endothelial cells), rat monoclonal CD31 antibody (1/100, #553370, RRID:AB_394816, BD Biosciences, Franklin Lakes, NJ, USA) was specific to blood vessels, mouse monoclonal E-Cadherin antibody (1/200, #610182, RRID:AB_397581, BD Biosciences) labeled the epithelial cells, and the rabbit polyclonal AIM2 antibody (1/100, #63660, RRID:AB_2890193, Cell Signaling Technology, Danvers, MA, USA) was used to detect AIM2-inflammasomes. 

Appropriate fluorochrome-conjugated secondary antibodies were obtained from Invitrogen (Waltham, MA, USA) and nuclei were counterstained with DAPI. The formation of inflammasome complexes following bacterial and sterile stimuli in WT mice and during chronic inflammation in TSP-1^-/-^ mice was detected using the ASC-citrine fusion protein that is constitutively and ubiquitously expressed in *R26^ASC-citrine^* mice and forms fluorescent specks upon inflammation. Images were taken using a LSM 880 laser scanning confocal microscope (Zeiss, Oberkochen, Germany) at the microscopy core of the Scripps Research Institute (La Jolla, CA, USA). Three different fields were analyzed per animal. Inflammasomes were counted using Imaris Spots software, which allows for the detection and counting of small particles. The number of specks detected in the inflamed LGs was adjusted by subtracting the number of specks found in their respective controls.

### 2.5. Western Blotting Analysis

For protein extraction, the dissected LGs were rinsed in cold PBS and transferred into 2 mL tubes pre-filled with 2.8 mm ceramic beads (#19-628, Omni, Inc., Kennesaw, GA, USA) containing 500 µL of ice-cold RIPA buffer without detergents (50 mM Tris-HCl + 150 mM NaCl + 1 mM EGTA) and supplemented with protease/phosphatase inhibitor cocktail (#5872, Cell Signaling, Danvers, MA, USA). The tissue was homogenized with the Omni Bead Ruptor 4 (# 25-010, Omni, Inc., Kennesaw, GA, USA; 2 cycles: Speed 5, 40 s each) and lysate was kept on ice. An appropriate volume of complete RIPA buffer supplemented with each detergent five-times concentrated was added to the sample (final concentration: 1% Nonidet P-40 + 0.5% sodium deoxycholate + 0.1% SDS + 1 mM EDTA) before incubation on ice for at least 30 min. Then, lysate was centrifuged (15 min, 15,000× *g*, 4 °C) and the supernatant was collected for protein quantitation using Pierce BCA protein Assay kit (#23225, Thermo Fisher Scientific, Waltham, MA, USA). After denaturation (with NuPAGE LDS + β-mercaptoethanol at 70 °C for 10 min), 30 µg total protein from each sample was separated by sodium dodecyl sulfate–polyacrylamide gel electrophoresis (NuPAGE Novex Bis Tris gels, ThermoFisher Scientific, Waltham, MA, USA) and transferred to polyvinylidene difluoride membranes using the iBlot2 system (ThermoFisher Scientific, Waltham, MA, USA). The transfer membranes were stained for total protein with the No-Stain Protein Labeling Reagent (#A44449, ThermoFisher Scientific, Waltham, MA, USA), according to the manufacturer’s instructions. The membranes were then blocked with TBS + 0.1% Tween + 5% milk for 1 h at RT before incubation with the appropriate primary antibody at 4 °C overnight: anti-GSDMDC (1/1000, #sc-393656, RRID:AB_2728694, Santa Cruz Biotechnology, Dallas, TX, USA,), anti-CASP1 (1/1000, #22915-1-AP, RRID:AB_2876874, ProteinTech, San Diego, CA, USA), or anti-IL-1β (1/1000, #A16288, RRID:AB_2769945, Abclonal, Woburn, MA, USA). After 3 washes, the blots were incubated for 1 h with the appropriate horseradish peroxidase-linked secondary antibody and processed for chemiluminescence. The signal was detected using the iBright (ThermoFisher Scientific, Waltham, MA, USA) or ChemiDoc MP (Bio-Rad, Hercules, CA, USA) imaging systems. The quantification of the relative protein abundance was performed using Image Lab software (v6.1, Bio-Rad, Hercules, CA, USA). Background-adjusted band volumes were corrected with the normalization factor (calculated using total protein stain for *BALBc/NOD.H2^b^* samples and β-actin for *TSP-1^-/-^* samples) and normalized to a reference volume corresponding to the average of the biological replicates in the control condition.

### 2.6. qRT-PCR

The total RNA was isolated using Trizol and converted into cDNA using RT^2^ First Strand Kit (#330404, Qiagen, Germantown, MD, USA). The gene expression was assessed using the SYBR Green kit, according to the manufacturer’s instructions (Applied Biosystems, Waltham, MA, USA) using a 7300 Real-Time PCR System (Applied Biosystems, Waltham, MA, USA). The sequences for qRT-PCR primers are listed in Table 1. qPCR data were analyzed using the comparative C_t_ (ΔΔC_t_) method. *Actb* and *Gapdh* were used as internal reference genes.

### 2.7. RNA-Sequencing (RNA-seq) Data Analysis

For acute injury experiments, we used the previously published data available from the Gene Expression Omnibus (GEO) database (Accession number: GSE99093) [13]. To study chronic inflammation, we mined our RNA-seq data of *BALBc* and *NOD.H2^b^* male LGs processed at 2, 4, and 6 months of age, deposited under GSE210332. Data were analyzed using ROSALIND^®®^ software (https://rosalind.bio/ accessed on 19 February 2023) developed by ROSALIND, Inc. (San Diego, CA, USA). As previously described, ROSALIND uses DESeq2 R library to normalize read counts and calculate fold-changes along with the corresponding *p*-values [37].

For all projects, differentially expressed genes (DEGs) were selected based on a fold-change (FC) cut-off equal to log_2_(FC) ≈ ±0.585 (corresponding to FC = 1.5) and *p*-adj < 0.05. On the figures, the statistical significance of log_2_(FC) compared to the respective control (uninjured LG for acute injury; age-matched *BALBc* for *NOD.H2^b^* LGs) is shown with: * *p*-value adj. < 0.05; ** *p*-value adj. < 0.01; *** *p*-value adj. < 0.001.

For acute injury data, a pathway enrichment analysis was conducted using Metascape [38] using default parameters (min. overlap = 3, *p*-value cut-off = 0.01, min. enrichment = 1.5) and Gene Prioritization by Evidence Counting (GPEC), an algorithm identifying the subset of input genes that are more likely to be true hits, and by interrogating WikiPathways and Gene Ontology Biological Process databases. The dendrogram was created with the web-based tool Clustergrammer (https://maayanlab.cloud/clustergrammer/ accessed on 25 May 2022) using complete-linkage clustering with Euclidian distances.

### 2.8. Statistical Analysis

Prism9 v9.1.2 (GraphPad software Inc, La Jolla, CA, USA) was used to plot the results and test their statistical significance. First, a Shapiro-Wilk normality test was performed to evaluate if the data follow a normal distribution. If data passed the normality test, the statistical significance between the two conditions was assessed with an unpaired *t*-test and the results were represented as mean ± standard deviation (SD). To compare more than two groups, a one-way ANOVA was used. Otherwise, non-parametric tests were used (Mann-Whitney test for two groups, and Kruskal-Wallis otherwise) and the plots showed the median ± interquartile range (IQR), as specified in the legends. The significant differences are represented as * if *p* value *p* < 0.05, ** if *p* < 0.01, and *** if *p* < 0.001.

## 3. Results

### 3.1. Epithelial Cells Can Sense Microbial/Sterile Inflammatory Stimuli

In contrast to non-canonical inflammasomes formed by CASP4, most types of canonical inflammasomes require the assembly of the adaptor protein ASC for the activation of the inflammasome cascade leading to IL-1β/IL-18 maturation, and eventually release through the GSDMD pores, also leading to cell death (Figure 1A). Thus, for the in vivo detection of the canonical inflammasome complexes, we used a transgenic mouse constitutively expressing the ASC-citrine fusion protein (*R26^ASC-citrine^*) (see enlarged micrograph in Figure 1A) [36]. To test whether inflammasomes could be formed in the LG, we first mimicked a bacterial infection in the *R26^ASC-citrine^* reporter mouse, as previously described [36]. In brief, LGs were injected with lipopolysaccharide (LPS, typical pathogen-associated molecular pattern, or PAMP), which constitutes the priming signal inducing *Il1b* transcription, and with nigericin, a pore-forming bacterial toxin which activates the NLRP3 inflammasome (Figure 1A) [39]. An anti-α-SMA antibody was used to identify MECs that surround the secretory units formed by acinar cells and the pericytes wrapped around blood vessels (blood vessels were labeled by antibody to CD31). Therefore, the tubular structures negative for CD31 and α-SMA staining were identified as ducts. In control LGs, we observed just a few ASC specks (Figure 1B and Appendix A). By contrast, numerous ASC specks were formed 6 h after LPS-nigericin stimulation, compared to the vehicle-injected LGs (Appendix A). Many of the ASC specks were found in the LG epithelium (Figure 1C and Appendix A). 

To test whether LG cells could also form inflammasomes upon sterile stimuli, we induced acute injury by injecting IL-1α in the LG on one side of the *R26^ASC-citrine^* mouse and injected the other LG with a saline (vehicle control) (Figure 1D,E). Similar to LPS-nigericin stimulation, ASC specks were detected 6 h after IL-1α-injection (Figure 1E). They were primarily formed in acinar cells and MECs (Figure 1E,F and Appendix A). At 12 h after injury, ASC complexes were also detected within ducts and infiltrating immune cells (Figure 1G). The proportion of cells forming inflammasomes and the diameter of ASC specks significantly increased over time in injured LGs (Figure 1H,I). Following inflammasome formation (Figure 1J), we also observed higher cell death rates with characteristic nuclear fragmentation in IL-1α-injected LGs (Figure 1K,L). 

Altogether, these results show that LG epithelial cells form inflammasomes in response to microbial or sterile pro-inflammatory stimuli and may eventually undergo pyroptosis prior to immune infiltration.

### 3.2. Inflammasome Activation during Chronic Inflammation

We then tested whether inflammasomes are activated in the LG epithelial cells during chronic inflammation. First, we analyzed *NOD.H2^b^* males, which develop robust lymphocytic infiltrates in the LG at 4–6 months of age [37] but do not have autoimmune diabetes [40,41]. To determine whether inflammasomes are active, we performed Western blotting to study the proteolytic maturation of downstream targets in the LGs of 6-month-old (6M) *NOD.H2^b^* and control *BALBc* males (Figure 2A,B and Appendix A). Although there was no significant difference in the abundance of the full length CASP1 (Pro-CASP1, p46) between the LGs of *NOD.H2^b^* and *BALBc* mice, it was detected at a slightly higher molecular weight in *NOD.H2^b^* mice (Figure 2A). This suggests a post-translational modification (ubiquitination, phosphorylation) that could modulate the inflammasome activity [42]. Moreover, in *NOD.H2^b^* LGs, we detected an increased amount of the cleaved form of CASP1 p33, which forms the active species on the inflammasome hub with the p10 subunit of CASP1 (Figure 2A). This complex is able to process many different substrates [43]. Consistent with inflammasome/CASP1 activation, we found that both the precursor (31 kDa) and the mature form of IL-1β (17 kDa) were significantly increased in the diseased LGs, and that the cleaved form of GSDMD was approximately five times more abundant in *NOD.H2^b^* as compared to *BALBc* LGs (Figure 2B). 

To determine if the activation of inflammasome signaling is not specific to *NOD.H2^b^* mice, we also analyzed the TSP-1-null (TSP-1^-/-^) mouse—another model of pSS [44,45]. We generated the *TSP-1^-/-^:R26^ASC-citrine^* mice and used the *R26^ASC-citrine^* mice as controls. While the control mice had only a few ASC specks in the LG, we found significantly more ASC specks in epithelial cells of the *TSP-1^-/-^*:*R26^ASC-^*^citrine^ mice at 2M and 6M, respectively (Figure 2C–F). The number of ASC specks increased with disease progression in the LGs of *TSP-1^-/-^*:*R26^ASC-citrine^* mice (Figure 2G) and correlated with the increased abundance of pro-IL-1β and IL-1β (Figure 2H). 

In summation, these results show that inflammasome complexes are constitutively formed in the LG of two murine pSS models of different background strains. Inflammasome activation gradually increases with the progression of the disease, suggesting that it could play a significant role in the pathogenesis through the secretion of inflammatory cytokines. 

### 3.3. Acute Injury and Chronic Inflammation Upregulate Several Types of Inflammasomes

Key components of the inflammasome machinery are highly expressed during inflammation in different tissues [24,46,47,48]. To identify the signaling pathways promoting their transcription, we analyzed a previously published RNA-seq data (GSE99093) obtained for LG acute injury/regeneration in *BALBc* mice [13]. Similar to our experiments, the LGs of these mice were injured by the intraglandular injection of IL-1α and the bulk RNA-seq was performed at days 0 (uninjected control), 1, 2, 3, 4, 5, 7, and 14 after injury. 

Consistent with our previous study [14], we found a strong induction of *Il1b* and *Il18* transcription on days 1 and 2 (Figure 3A, Appendix A). Therefore, we investigated the expression of all genes listed in the Gene Ontology Biological Process entitled “Interleukin-1 production” (GO:0032612) that were significantly enriched on days 1 and 2 after injury (Figure 3B, Appendix A). Thus, the gene expression heatmap demonstrates that LG samples from day 1 and 2 after injury cluster together, showing an upregulation of many genes of this pathway, while data from day 3 after injury display a significant decrease in the set of genes involved in inflammasome activation and interleukin processing (Figure 3B). In agreement with the original study [13], uninjected (day 0) and saline-injected controls (day “14S”) were highly similar (Figure 3B) and, thus, we retained the uninjected LGs as controls to study the gene expression changes. The main cluster of genes upregulated during the inflammatory phase (lower part of the heatmap) included the genes of the toll-like receptor (TLR)-myeloid differentiation primary response 88 (MyD88)-nuclear factor kappa B (NFκB) axis that promotes the transcription of *Il1b* and inflammasome components [49,50], and its partners such as *F2rl1* that acts synergistically with TLR2 and TLR4 [51,52], both upregulated in this dataset (see selected genes shown in frame on Figure 3B, Appendix A). The inflammasome sensors *Ifi204* (murine ortholog of human *IFI16* [49]), *Nlrp3*, *Aim2*, *Mefv* (encoding PYRIN), and *Naip5* (that forms hetero-oligomeric inflammasomes with NLRC4 upon recognition of bacterial fragments [53]) were significantly upregulated on days 1 and/or 2 after acute injury, and decreased to basal levels on day 3 (Figure 3B,C, Appendix A). By contrast, there was no significant change in the expression of *Nlrp6*, *Nlrc4*, and *Nlrp1b* (Figure 3C)*. Nlrp1a* was not expressed at any time (Figure 3C), which was expected, as the *BALBc* strain lacks *Nlrp1a* expression [54]. *Casp1* and *Casp4* were significantly increased during the first two days after injury, while the expression of both caspases returned to basal levels on day 3 after injury (Figure 3B and Appendix A). There was a modest increase in the *Casp4* mRNA level at days 4 and 5 but, from day 7, its expression level was not different from the uninjected controls. In addition, we detected the upregulation of the NLRP3-activator *Gbp5*, and the mediators of pyroptosis *Panx1* and *Gsdmd* (Figure 3B, Appendix A). 

The results obtained by qRT-PCR performed in our lab after acute injury of LG in *C57BL/6J* mice were highly similar to the RNA-seq data from *BALBc* mice (Appendix A). This demonstrates that the experimental model of LG acute injury provides mechanistically robust and reproducible results, even when different mouse strains are used. Therefore, these findings suggest that several types of canonical inflammasomes (NLRP3, AIM2, IFI204, PYRIN, NAIP5/NLRC4, and possibly non-canonical inflammasomes) are transiently activated by IL-1α-injection. 

We also analyzed the transcriptomic changes leading to inflammasome priming and activation during the development of chronic inflammation. To do this, we mined our previously published RNAseq data of LGs from 2M, 4M, and 6M *NOD.H2^b^* (diseased) and *BALBc* (control) males (GSE210332) [37]. In this study, we showed that, although there is a major shift in the gene expression between 2M and 4M/6M mice, 2M *NOD.H2^b^* LG already features many alterations at the transcriptomic level compared to *BALBc* controls. We also found that B and T cell infiltrates appeared as early as 2M (early stage of the disease) in *NOD.H2^b^* males—although not to the same extent as 6M males (clinical stage).

Consistent with our previous observations, the expression of *Il1b* and *Il18* was significantly increased with the disease progression (Figure 4A) and genes for the “Interleukin-1 production” (GO:0032612) pathway were significantly enriched in the list of differentially expressed genes (DEGs) between all diseased and control animals. Our data show a set of genes that are upregulated in *NOD.H2^b^* mice (Figure 4B, Appendix A). Among these genes, we found several pro-inflammatory factors, including TLRs, *Tnf,* and *Nod2*, as well as *Ifng,* (coding for IFN-γ) that participate in NF-κB activation (Figure 4B and Appendix A). IFN-γ not only promotes the transcription of inflammasome components, but also their assembly through the activation of GBPs [55,56]. In *NOD.H2^b^* mice, the expression of genes for inflammasome sensors *Nlrp3*, *Nlrc4*, *Naip5*, *Ifi204*, *Aim2*, and *Mefv* were significantly increased from 2 months of age, while no significant changes were detected for *Nlrp1b* and *Nlrp6* mRNAs (Figure 4C, Appendix A). Although *Nlrp1a* is not expressed in *BALBc* mice, we noticed that its mRNA expression significantly increased with age in *NOD.H2^b^* LGs. Importantly, the upregulation of inflammasome sensors was associated with the increased transcription of *Casp1* and *Casp4* (Figure 4B and Appendix A), along with *Gsdmd* and *Panx1* (Figure 4B, Appendix A). While most of inflammasome-related genes were significantly upregulated at 2M compared to the healthy controls, the expression level of some of them (including *Aim2*, *Nlrc4*, *Nlrp1a*, *Casp1*, *Panx1*, and *Gsdmd*) increased further in *NOD.H2^b^* LGs at 4M (Appendix A). By contrast, none of the genes from the “Interleukin-1 production” pathway passed our thresholds when comparing 4M and 6M *NOD.H2^b^* LGs. No DEGs were found for the 2M/4M and 4M/6M comparisons in the control *BALBc* LGs. This shows that, in *NOD.H2^b^* LGs, the activation of the inflammasome pathway amplifies up to 4M, the age at which the chronic inflammation is established [37].

Therefore, during chronic inflammation, several transcriptional activators of inflammasome components are upregulated from the early stages of the disease. Compared to acute injury, which leads to the upregulation of *Nlrp3*, *Aim2*, *Ifi204*, *Mefv*, and *Naip5*, chronic inflammation also increased *Nlcr4* expression, suggesting that most of the identified sensors can be involved in both acute and chronic inflammation. By immunostaining, we confirmed an increased signal for AIM2 expression in *NOD.H2^b^* compared to *BALBc* LGs (Appendix A). In *NOD.H2^b^* LGs, AIM2 complexes were found in the perinuclear areas of epithelial cells, and eventually in other compartments, such as blood vessels (Appendix A). The constitutive activation of inflammasomes in *NOD.H2^b^* LGs could be also facilitated by the upregulation of the gene encoding SYK (spleen-associated tyrosine kinase) (Appendix A), which induces ASC phosphorylation and oligomerization that is essential for the assembly of NLRP3 inflammasome and CASP1 activation [57]. Finally, the upregulation of *Casp4* and members of IFN-γ signaling suggests the activation of the non-canonical inflammasomes.

Taken together, these results demonstrate that the expression/activation of multiple inflammasome complexes in the LG is transient during acute inflammation but is sustained during chronic inflammation. Both acute and chronic inflammasome activities likely lead to epithelial cell damage, the induction of adaptive immune response, and the formation of lymphocytic infiltrates. Therefore, we investigated whether molecular pathways promoting the resolution of inflammation and inflammasome inhibition after acute injury are altered during chronic inflammation.

### 3.4. The Resolution of Inflammation and LG Regeneration following Acute Injury Is Concomitant with the Activation of Lipid Metabolism

An analysis of the transcriptome of regenerating LGs determined day 3 as the critical time point for the switch between inflammatory and regenerative processes [13]. This time point is characterized by a significant decrease in the CD45^+^ (immune) cells in the LG, particularly neutrophils and monocytes as previously reported [13], and a decrease in the inflammasome components and a reduction in *Il1b* and *Il18* expression down to basal levels. Therefore, we hypothesized that the genes involved in the resolution of inflammation and inhibiting the inflammasome signaling are upregulated on day 3 after LG injury. 

We identified 337 differentially expressed genes (DEGs) on day 3 after the injury (relative to the uninjured control, log_2_(FC) cutoff = ±log_2_(1.5) and *p*-adj < 0.05), including 219 upregulated genes. An analysis of these 219 upregulated genes with Metascape demonstrated that most of the top 30 significant biological pathways were related to the activity of immune cells and inflammatory responses (Figure 5A). The protein-protein interaction enrichment analysis carried out with these 219 genes (Appendix A) identified 6 densely connected network components related to: (1) phagocytosis and tumor necrosis factor (TNF) production, (2) mitosis, (3) sterol biosynthesis, (4) leukocyte/myeloid activation, (5) inflammatory response, and (6) memory. The heatmap comparing the enrichment *p*-values between the first three days after injury revealed that the inflammatory pathways belong to the primary response highly activated on days 1 and 2. Indeed, these pathways become much less significant on day 3, meaning that many of these genes were not upregulated anymore (Figure 5A). Hierarchical clustering showed a group of pathways related to the metabolism and localization of lipids that were significantly enriched on day 3, but not significant on days 1 and 2. Nonetheless, some of the genes that were involved in sterol biosynthesis (*Acat2*), inert fat breakdown (*Lipa*), and lipid transport/signaling (*Abcg1*, *Apobec1*, *Apoe*, *Lrp1*, *Ldlr*, *Pltp*, *Ptafr*) were maximally expressed on day 2—except for *Ldlr*, whose expression level reached a peak on day 3 (Appendix A). 

To exclude the genes belonging to the primary inflammatory response initiated on day 1 and to identify the mechanism(s) specifically promoting the switch to LG repair on day 3 after the injury, we selected genes specifically upregulated on day 3 after the LG injury (53 genes) and genes upregulated on day 2 with the expression level maintained or increased from day 2 to day 3 (13 genes) (Appendix A). Based on these criteria, we generated a list of 66 genes (Appendix A) and performed gene set enrichment analysis with Metascape to interrogate several ontology sources (Gene Ontology, KEGG Pathway, Reactome, WikiPathways) (Figure 5B). The enriched ontology terms were grouped by similarity into clusters and named after the most significant pathway. Most of the resulting biological processes were interconnected and related to the biosynthesis and metabolism of lipids (Figure 5C). The most significant cluster with 18 differentially activated genes was “Cholesterol metabolism with Bloch and Kandutsch-Russell pathways” (Figure 5B). Among these genes, we identified *Srebf1* reaching its highest level on day 3 after IL-1α injury (Figure 6A), and its inhibitors *Insig1* and *Prkaa2* (coding for a subunit of AMPK) (Appendix A) [58,59]. *Srebf1* encodes the transcription factor sterol regulatory element-binding protein 1 (SREBP-1), a master regulator of lipid homeostasis (Figure 6B). The translocation of SREBP-1 into the nucleus activates the transcription of genes regulating the biosynthesis and uptake of fatty acids and cholesterol [59] (Figure 6B). Consistent with SREBP-1 activation, many enzymes of the pathways processing acetyl-CoA for the biosynthesis of fatty acids and cholesterol reached their highest expression level on day 3 after injury (Appendix A), including the rate-limiting enzymes acetyl-CoA carboxylase beta (ACACB) and squalene epoxidase (SQLE) (Figure 6A,B). Similarly, the gene encoding Acyl-CoA synthetase short-chain family member 2 (ACSS2) that catalyzes the formation of acetyl-CoA and the activation of fatty acids into fatty acyl-CoA (Figure 6B) was upregulated only on day 3 (Figure 6A). We also found an upregulation of the predicted lipase gene *Gm8978* and other lipid-related genes: *Aldh3b2*, coding for the aldehyde dehydrogenase 3 family member B2 (ALDH3B2) protein that removes toxic aldehydes from lipid droplets, and *Dgkg*, encoding diacylglycerol kinase gamma (DGKγ), which produces phosphatidic acid (PA) by phosphorylating the second messenger diacylglycerol (DAG) (Appendix A). 

Taken together, these results confirmed that the inflammatory pathways significantly enriched among the 219 upregulated DEGs outlined the primary immune response that occurred within days 1 and 2. Furthermore, in contrast to cholesterol conversion and transport pathways that increased quickly after injury, most of the genes involved in lipid biosynthesis were upregulated only on day 3, suggesting that this transcriptional program may specifically control the resolution of inflammation and gland regeneration. The activation of lipid metabolism most likely promotes the synthesis of cell membranes (that is composed of triglycerides, phospholipids, cholesterol), adenosine triphosphate (ATP) production (through β-oxidation and the use of acetyl-CoA by the TCA cycle), and the generation of second messengers (i.e., PA and DAG) regulating various cell processes (Figure 6A), and may also repress the inflammasome pathway. 

### 3.5. Lipid Metabolism Is Altered in Chronically Inflamed LG of NOD.H2^b^ Mice

We hypothesized that the mechanism(s) promoting the resolution of inflammation after acute injury might be altered in *NOD.H2^b^* mice. We have recently reported that the downregulation of genes involved in fatty acid biosynthesis, TCA cycle, and fatty acid β-oxidation was associated with disease progression in *NOD.H2^b^* LGs [37]. Since these pathways are interconnected and participate in lipid homeostasis (Figure 6B), we analyzed the genes of relevant pathways that were significantly enriched in the meta-analysis of *BALBc/NOD.H2^b^* comparisons, according to Gene Ontology (*p*-elim < 0.05) and WikiPathways (*p*-adj < 0.05) (Appendix A).

Among these genes, we found key regulators of lipid metabolism downregulated in the LG of *NOD.H2^b^* mice as early as 2 months of age (Appendix A). Most altered were the genes coding for peroxisome proliferator-activated receptor alpha (PPARα, encoded by *Ppara*, Figure 7A) and its partner retinoid X receptor alpha (RXRα, *Rxra)* that regulate fatty acid transport, oxidation ketogenesis, and promote SREBP-1 activity [60,61]. There was also a downregulation of *Agt* coding for the angiotensin precursor ANGT that promotes the transcription of lipogenic genes, such as *Srebf1. Srebf1* itself was reduced (Figure 7A) along with its synergic partner ChREBP (encoded by *Mlxipl*, Appendix A) that, together with SREBP-1, activates the transcription of genes involved in fatty acid biosynthesis. 

Consistent with the downregulation of the PPARα-dependent transcriptional program, the expression levels of enzymes involved in mitochondrial fatty acid β-oxidation were also reduced in the LGs of the *NOD.H2^b^* mice (Appendix A, Figure 7B). Similarly, we noted the downregulation of enzymes mediating ketone catabolism such as *Oxct1* (Appendix A). Altogether, this suggests a reduced mitochondrial pool of acetyl-CoA available for the TCA cycle. The decreased TCA cycle activity most likely reduces ATP production and also the amount of citrate that can be used for acetyl-CoA synthesis in the cytosol (Figure 6A). We also found the downregulation of enzymes catalyzing the synthesis of acetyl-CoA from citrate, acetate, or pyruvate (Appendix A), including ACSS2 (Figure 7C), which also activates fatty acids by the reaction with acetyl-CoA to form fatty acyl-CoA (Figure 6A). 

Since acetyl-CoA is the primary substrate for de novo lipid biogenesis, we hypothesized that this process is altered in chronically inflamed LGs. In addition to reduced *Srebf1* levels, many enzymes for fatty acid biosynthesis such as *Acacb* were indeed downregulated in diseased glands, thus suggesting a reduced generation of free fatty acids and activated acyl-CoA (Figure 7C,D and Figure 8, Appendix A). The number of significant DEGs in *NOD.H2^b^* LGs and the extent of the alterations in their expression level compared to their respective *BALBc* controls increased with age, especially between 2M and 4M (Figure 7C,D, Appendix A). By contrast, the enzymes catalyzing the elongation of long-chain fatty acids (LCFAs) into very long-chain fatty acids (VLCFAs) (*Elovl5*, *Elovl6*, *Hacd4)* were upregulated in diseased LGs (Figure 7D, Appendix A). VLCFAs serve as precursors for eicosanoids such as inflammatory prostaglandins, thus suggesting that eicosanoid metabolism was also altered in *NOD.H2^b^* mice. 

In addition, many enzymes of the pathways leading to cholesterol synthesis were upregulated during chronic inflammation, including the rate-limiting enzymes 3-Hydroxy-3-Methylglutaryl-CoA Reductase (HMGCR) and SQLE (Figure 7E,F, Appendix A). We also noticed the downregulation of *Cyp27a1* and *Cyp46a*, which convert free cholesterol into secreted metabolites. Finally, there was a significant increase in the expression level of *Soat1* and *Soat2* catalyzing the formation of cholesterol esters (Figure 7E,F, Appendix A). Of note, the expression level of *Srebf2*, which preferentially promotes cholesterol biosynthesis, was not altered by chronic inflammation (Appendix A). This supports our observation that fatty acids, but not cholesterol synthesis, are reduced during chronic inflammation.

Some of the genes altered during chronic inflammation were in the list of lipogenic genes upregulated on day 3 after injury so we analyzed the expression of the entire gene set activated at the beginning of the regenerative phase after acute injury (Appendix A). As expected, we found a significant downregulation of enzymes responsible for acetyl-CoA and fatty acid biosynthesis contrasting with the upregulation of genes related to cholesterol biosynthesis, transport, and lipases. 

All these observations indicate an alteration of mitochondrial metabolism—with decreased acetyl-CoA and ATP production—and of de novo lipid biosynthesis using these metabolites, due to the downregulation of PPARα/SREBP-1-dependent transcriptional programs (Figure 8). It is possible that enzymes involved in eicosanoid metabolism might promote the synthesis of pro-inflammatory lipids over anti-inflammatory metabolites (Appendix A, Figure 8). Considering the overall upregulation of enzymes of the mevalonate and lanosterol pathways (Figure 8), the chronically inflamed glands may accumulate cholesterol, as previously reported in *NOD* mice [62]. Together with the impaired mitochondrial and fatty acid metabolism, this may induce epithelial cell stress and promote inflammasome activation in the LGs of *NOD.H2b* mice. Few of these alterations were significantly aggravated between 2M and 4M/6M *NOD.H2^b^* mice (Appendix A). Only *Acacb* was further downregulated at each time-point, while it was not affected in *BALBc* mice. Altogether, this suggests that changes in lipid metabolism are one of the earliest mechanisms of disease development. 

## 4. Discussion

In this study, we showed that several types of inflammasomes could be activated in LG epithelial cells during acute and chronic inflammation. This suggests that, similar to corneal epithelial cells [63], LG epithelial cells function as sentinel cells [64]. We also found that inflammasome activation precedes epithelial cell death after IL-1α-induced acute injury and discovered increased GSDMD cleavage in chronically inflamed LGs of *NOD.H2^b^* mice. Studies in the salivary gland suggest that inflammasome activation and pyroptosis exacerbate inflammation through immune cell infiltrations in SS patients and promote salivary gland dysfunction [28,29]. To our knowledge, there is no such report about pyroptotic events in the lacrimal gland. However, topical administration of anakinra (IL-1 receptor antagonist) to the cornea improved the ocular surface integrity and tear secretion in Aire-deficient mice with SS-like disease [65], suggesting that secretion of IL-1β by epithelial LG cells during chronic inflammation could participate in corneal damage. 

During acute and chronic inflammation, various types of inflammasome sensors are upregulated in the LG, thereby illustrating the complexity of the innate inflammatory response in this organ. Surprisingly, both acute and chronic inflammation activated similar inflammasome sensors: (1) AIM2 and IFI204 that are activated by DNA released from damaged mitochondria or dead cells; (2) NLRP3 that is activated by a wide variety of stimuli, including oxidative stress and stress-induced lipid signaling; (3) PYRIN, whose loss-of-function mutation causes the monogenic autoinflammatory disease familial Mediterranean fever [66]; and (4) NLRC4 that, if mutated, causes constitutive CASP1 cleavage in cells leading to severe autoinflammatory syndromes in humans [67]. In the LG, *Nlrc4* was upregulated only by chronic inflammation but the transcription of its partner NAIP5 was increased during both acute and chronic inflammation. Our data also show that several signaling pathways essential for NF-κB/inflammasome activation, including TLR/MyD88, were upregulated by acute and chronic inflammation. MyD88-deficiency was shown to significantly dampen disease development in *NOD.H2^b^* mice [68], thus supporting a critical role in SS pathogenesis for this signaling pathway. In our study, we also noted the robust upregulation of GBPs and SYK that were recently described as modulators of inflammasome signaling [69]. SYK controls the activation of AIM2 and NLRP3 inflammasomes by phosphorylating ASC [70], thereby promoting its oligomerization and the recruitment of pro-CASP1 [71]. GBPs are part of the interferon signature that is involved in the pathogenesis of SS [72] and may play a role in SS and other autoimmune diseases by regulating inflammasome activation. Taken together, we propose that the sustained activation of inflammasome pathways observed in *NOD.H2^b^* and *TSP-1^-/-^* mice contribute to LG chronic inflammation. 

In the search for molecular suppressors of inflammasome signaling following IL-1α injection, we discovered that *Srebf1* and the genes involved in lipid biosynthesis/transport were upregulated at the resolution of inflammation. As for now, the link between lipid metabolism and inflammasome activity was mostly studied in immune cells. During the acute phase of inflammation, macrophages accumulate cholesterol, which activates inflammasomes through TLR signaling [73]; for example, by forming TLR4-inflammarafts that upregulate *Il1b* as shown in microglia [74]. Oishi and co-authors elegantly showed that in macrophages, SREBP-1 is first inhibited and then induced by TLR4/MyD88 at the later stages of the inflammatory response to promote the synthesis of anti-inflammatory fatty acids that promote the resolution of inflammation [75]. Similarly, the SREBP-dependent lipogenic program is induced by the activation of CASP1 by NLRP3 and NLRC4 inflammasomes in vitro [76]. Whether similar crosstalk between SREBP-1 signaling and inflammasomes occurs in LG epithelial cells remains to be determined. 

By contrast, we showed that reduced mitochondrial metabolism (fatty β-oxidation, TCA cycle) and fatty acid metabolism were associated with disease progression in *NOD.H2^b^* mice and mainly affected LG epithelium [37]. *NOD.H2^b^* LGs might display similar mitochondrial alterations as diabetic *NOD* mice [77] and the mitochondrial damage itself could enhance inflammasome signaling [78]. A decrease in acetyl-CoA pools may also have a profound effect on gene expression through protein acetylation [79,80] and impair de novo lipid biosynthesis. Our data indeed showed the downregulation of PPARα/SREBP-1 signaling and of downstream genes involved in fatty acid metabolism, while genes promoting the generation and transport of cholesterol were upregulated at the early stages of the disease. Thus, the activation of cholesterol biosynthesis could be SREBP-1-independent and/or higher free cholesterol levels would exert negative feedback on *Srebf1* expression. We also found the downregulation of *Cyp27a1* and *Cyp47a1* catalyzing the conversion of cholesterol into the secreted form 25-Hydroxycholesterol that reduces *Il1b* transcription and CASP1 activation [81]. 

In agreement with our study, Wu et al. (2009) showed altered lipid homeostasis in the LGs of diabetic *NOD* males [62]. In this model, cholesterol ester accumulation preceded lymphocytic infiltration and was not a consequence of dacryoadenitis. Altered lipid homeostasis is not restricted to mouse models of SS, since fat deposition in the LG is a feature of SS patients [82]. We previously reported large cytoplasmic vacuoles in LG acinar cells of *NOD.H2^b^* mice, suggesting they accumulate lipid droplets [37]. The activation of the lanosterol/mevalonate pathway could be a compensatory mechanism aimed at increasing other non-steroid products, for example, to rescue mitochondrial function [83]. The resulting cholesterol accumulation in epithelial cells might in turn downregulate SREBP-1-dependent lipid biosynthesis, promote lipotoxic damage, and activate inflammasome signaling. 

Consistent with our findings, PPARα is also downregulated in experimental mouse models of LG inflammation induced by high-fat diets [84] or obstructive sleep apnea [85] that lead to lipid accumulation and dry eye symptoms. The latter can be alleviated by fenofibrate [84,85], an FDA-approved PPARα activator. Fenofibrate is a hypolipidemic drug that is used to treat the symptoms of high cholesterol and triglycerides in human. Recently, Guo and co-authors showed that fenofibrate improved tear production and corneal surface state and reduced lymphocytic infiltrates in the LGs of *NOD/ShiLtJ* mice through the modulation of Th17/T_reg_ cell differentiation [86]. Thus, promoting fatty acid β-oxidation and biosynthesis through the activation of PPARα/SREBP-1 signaling could inhibit pro-inflammatory pathways in the LGs of *NOD.H2^b^* mice and promote regenerative processes. 

The crosstalk between epithelial cells and lymphocytes plays a key role in SS development [87,88]. In fact, anti-inflammatory drugs such as Rituximab (B-cell depleting agent) and Anakinra (anti-IL-1) had only transient or no effects on SS patients [87,89]. We thus believe that anti-inflammatory molecules, combined with drugs that restore epithelial cell homeostasis, may lead to better outcomes. Therefore, one possible avenue of research is the combination of fenofibrate with iguratimod (anti-rheumatic drug inhibiting TNF-α, IL-1, IL-6, BAFF-R, CD38 signaling) or necrosulfonamide (inhibitor of GSDMD-pore formation) to decrease inflammation, reduce deleterious effects of inflammasome activation, and durably improve the epithelial function in SS.

## 5. Conclusions

In summary, our work shows that, in addition to secreting antimicrobial and immunoregulatory factors into the tear fluid, the lacrimal gland epithelium plays a pivotal role in the innate immune response by activating inflammasome signaling in response to exogenous or endogenous stimuli. The dysregulation of this protective defense mechanism during chronic inflammation is associated with an imbalance between the metabolic pathways producing fatty acids and cholesterol. These alterations contribute to pSS pathogenesis and thus represent a new avenue for therapeutics development.

## Figures and Tables

**Figure 1 ijms-24-04309-f001:**
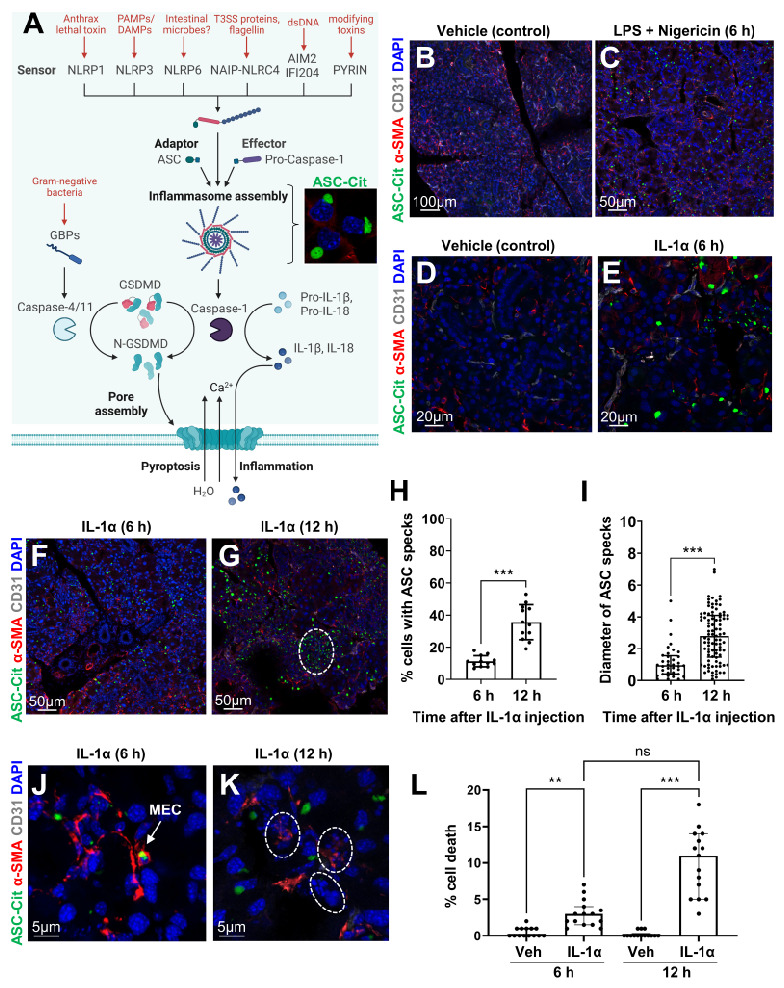
**Microbial and sterile stimuli induce the assembly of canonical inflammasome in LG epithelial cells.** (**A**) Overview of the inflammasome pathway. Depending on the PAMP/DAMPs (pathogen/damage-associated molecular patterns) detected, a specific inflammasome sensor will oligomerize and recruit the adaptor protein ASC (apoptosis-associated speck-like protein containing a CARD) and the effector pro-caspase-1 to form the canonical inflammasome complex. Inflammasome formation in the LG can thus be monitored in vivo using the *R26^ASC-citrine^* reporter mouse forming green, fluorescent ASC specks (see micrograph). After auto-activation, caspase-1 cleaves gasdermin D (GSDMD) and the precursors for interleukins IL-1β and IL-18. Guanylate-binding proteins (GBPs) activate caspase-4/11, which also cleaves GSDMD. The N-terminus of GSDMD then forms a pore at the plasma membrane, thus enabling the release of mature IL-1β and IL-18, and the entry of water and calcium. This induces pyroptosis through cell swelling and calcium-dependent activation of calpains. Created with BioRender.com. (**B**–**L**) LG sections were analyzed for ASC specks formation (green) and immunostained for α-SMA (red) and CD31 (grey) to visualize MECs and blood vessels, respectively. *R26^ASC-citrine^* mice (*n* = 3 per group) were injected with (**B**,**D**) vehicle control, (**C**) LPS + nigericin, or (**E**–**L**) IL-1α, and harvested at (**C**,**E**,**F**,**J**) 6 h or (**G**,**K**) 12 h after injury. (**G**) Dashed circle indicates immune infiltrates. (**H**) Percentage of cells displaying ASC specks after IL-1α injection (mean (SD), unpaired *t*-test). (**I**) Size of specks increased over time after IL-1α injection (median (IQR), Mann-Whitney test). (**K**) Dashed circles indicate nuclei fragmentation during cell death. (**L**) Proportion of dying cells after IL-1α injection (Median (IQR), Kruskal-Wallis test). Significant differences are represented as ** if *p* < 0.01, and *** if *p* < 0.001; “ns” indicates non-significant differences.

**Figure 2 ijms-24-04309-f002:**
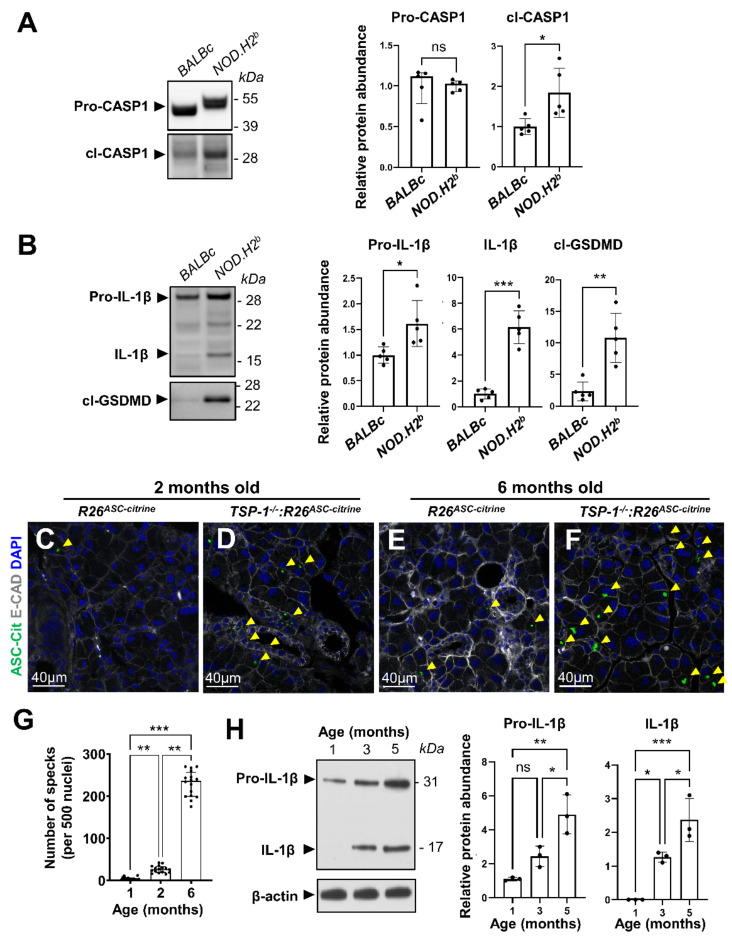
**Analysis of inflammasome activation in the LG of Sjögren’s syndrome mouse models**. (**A**,**B**) Representative results and quantification for Western blot analysis of (**A**) Pro-CASP-1 (median (IQR), Mann-Whitney test), cl-CASP1 and (**B**) its downstream targets IL-1 (precursor and mature cytokine), and cl-GSDMD (C-terminus) (mean (SD), unpaired *t*-test) relative abundance in the LGs of 6-month-old *BALBc* and *NOD.H2^b^* mice (*n* = 5 animals). Normalization factor was calculated based on total protein stain. (C-F) LGs from (**C**,**D**) 2- and (**E**,**F**) 6-month-old *R26^ASC-citrine^* (**C**,**E**) wild-type or (**D**,**F**) *TSP-1^-/-^* mice were immunostained for E-cadherin (E-CAD) to show ASC specks (yellow arrowheads) in epithelial cells. (**G**) Quantification of ASC-citrine specks in *TSP-1^-/-^-R26^ASC-citrine^* (adjusted to WT LG, median (IQR), Kruskal-Wallis test). (**H**) Western blot analysis and quantification of pro- and mature IL-1 in the LG of 1-, 3-, and 5-month-old *TSP-1^-/-^* mice (*n* = 3, mean (SD), one-way ANOVA). Significant differences are represented as * if *p* value *p* < 0.05, ** if *p* < 0.01, and *** if *p* < 0.001; “ns” indicates non-significant differences.

**Figure 3 ijms-24-04309-f003:**
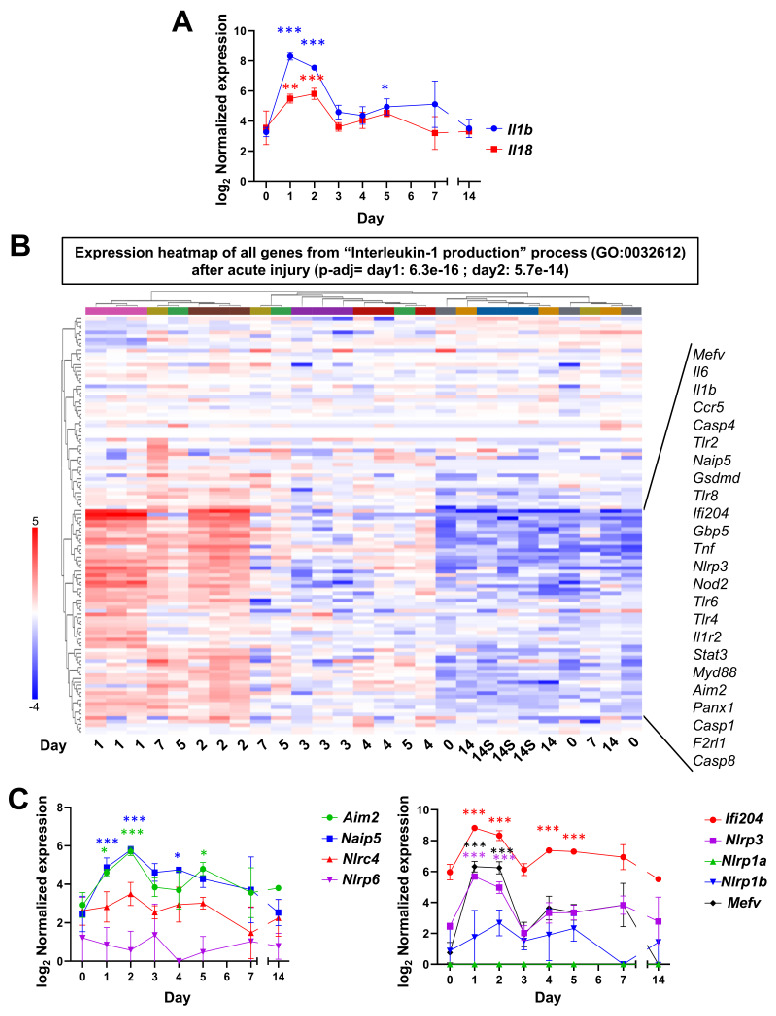
**RNA-seq analysis of inflammasome pathway after IL-1α-induced acute LG injury.** After IL-1α injection into the LG on day 0, (**A**) *Il1b* and *Il18* transcripts were upregulated on days 1 and 2 by RNA-seq. (**B**) Expression heatmap of the genes involved in the pathway entitled “interleukin-1 production” (GO:0032612) show that individual LG samples cluster in 3 main groups, with biological replicates from days 1 and 2 after injury, present an important activation of a set of genes involved in this pathway. Samples from days 3, 4 and 5 show an intermediate profile, while samples from days 14 (IL1-α-injected) and 14S (saline-injected) are similar to uninjected controls (day 0). Genes displayed in the frame to the right were selected from the cluster of upregulated genes as closely associated with inflammasome complexes or as main triggers for their transcription. (**C**) mRNA expression of the different inflammasome sensors in the LGs after injury with IL-1α. Error bars show the mean (SD). Significant differences are represented as * if *p* value *p* < 0.05, ** if *p* < 0.01, and *** if *p* < 0.001 and colored as the corresponding gene.

**Figure 4 ijms-24-04309-f004:**
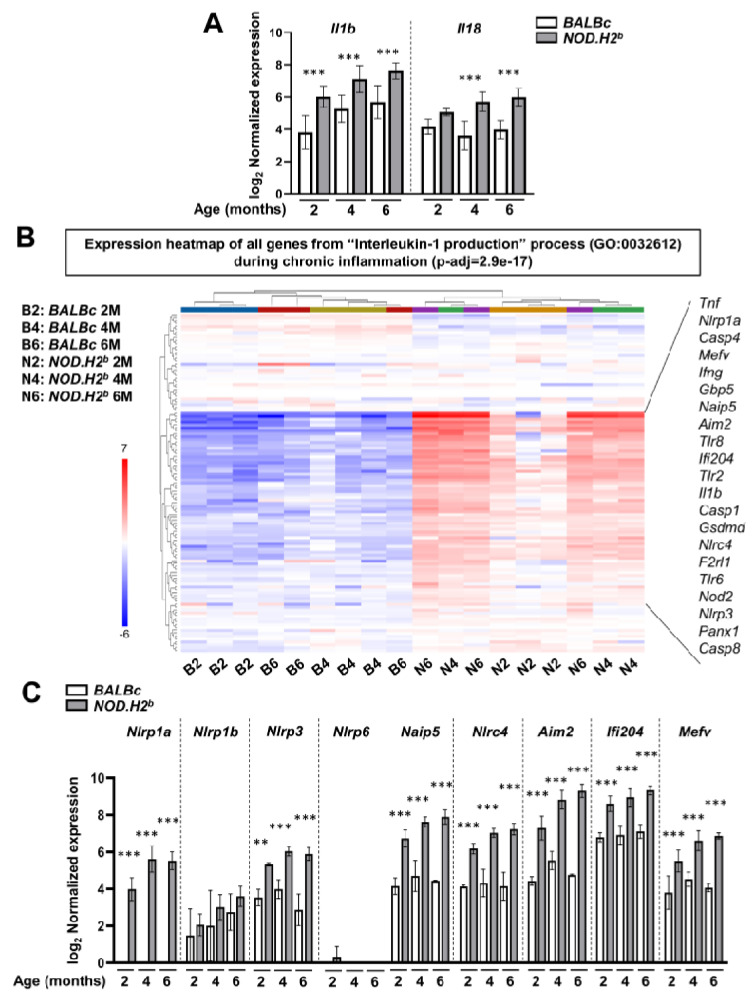
**RNA-seq analysis of inflammasome pathway during chronic inflammation.** (**A**) mRNA expression of *Il1b* and *Il18* is increased in *NOD.H2^b^* LGs (grey) compared to *BALBc* (white) at 2M, 4M, and 6M. (**B**) Expression heatmap of the genes involved in “interleukin-1 production” (GO:0032612) pathway in each biological replicate (1 sample = 1 LG) from *BALBc* (“B”) or *NOD.H2^b^* (“N”) mice at 2, 4, or 6M. Individual LG sample cluster in two main groups; all *NOD.H2^b^* samples show an important activation of genes involved in this pathway compared to *BALBc*. Selected genes that participate directly in inflammasome complexes/signaling or are upstream activators are shown in frame on the right. (**C**) mRNA expression of the different inflammasome sensors in *NOD.H2^b^* LGs (grey) compared to *BALBc* (white) at 2M, 4M, and 6M. Error bars show the mean (SD). Significant differences are represented as ** if *p* < 0.01, and *** if *p* < 0.001.

**Figure 5 ijms-24-04309-f005:**
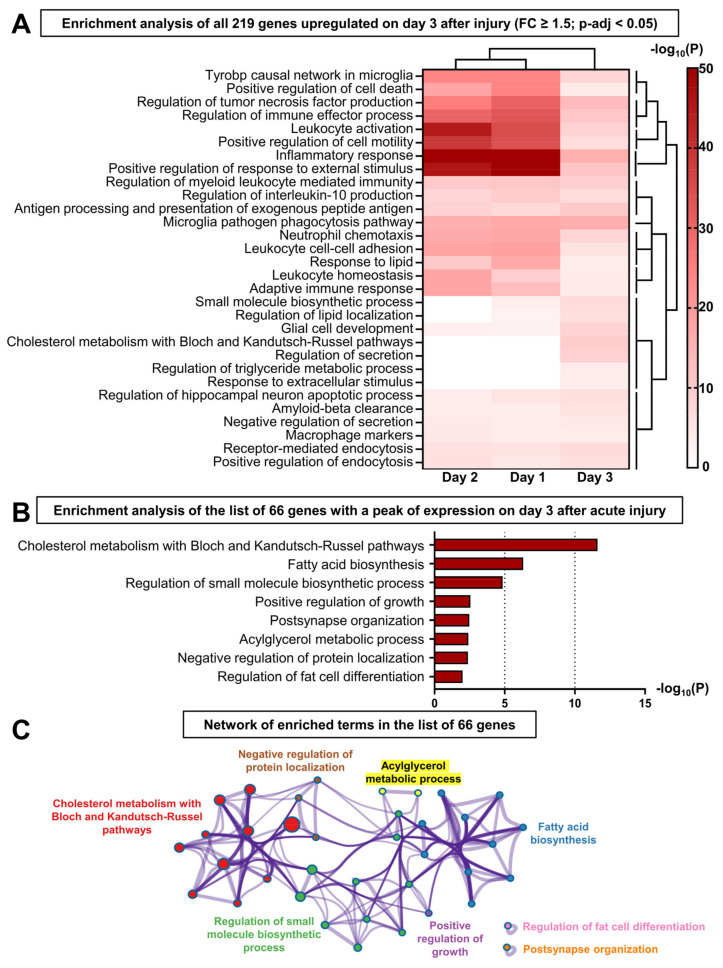
**Activation of lipid biosynthesis is specifically activated at the switch to repair after LG injury.** (**A**) Top 30 enriched pathways obtained by analyzing the list of 219 genes upregulated (relative to uninjected control, FC > 1.5, *p*-adj < 0.05) on day 3 after IL1-α-injury using Metascape. Enrichment *p*-values found on days 1 and 2 were added and a dendrogram was created with Clustergrammer to identify specific enrichment patterns. (**B**) Pathway enrichment analysis of the list of 66 genes supposedly promoting the resolution of inflammation using Metascape. (**C**) Network of enriched ontology terms obtained from the list of the 66 upregulated genes. One term from each cluster is selected to have its term description shown as cluster identity. Nodes that share the same cluster identity are typically close to each other and have the same color (i.e., nodes of the same color belong to the same cluster). The node size reflects the number of input genes that fall under that term. The thickness of the edge represents the similarity score (>0.3).

**Figure 6 ijms-24-04309-f006:**
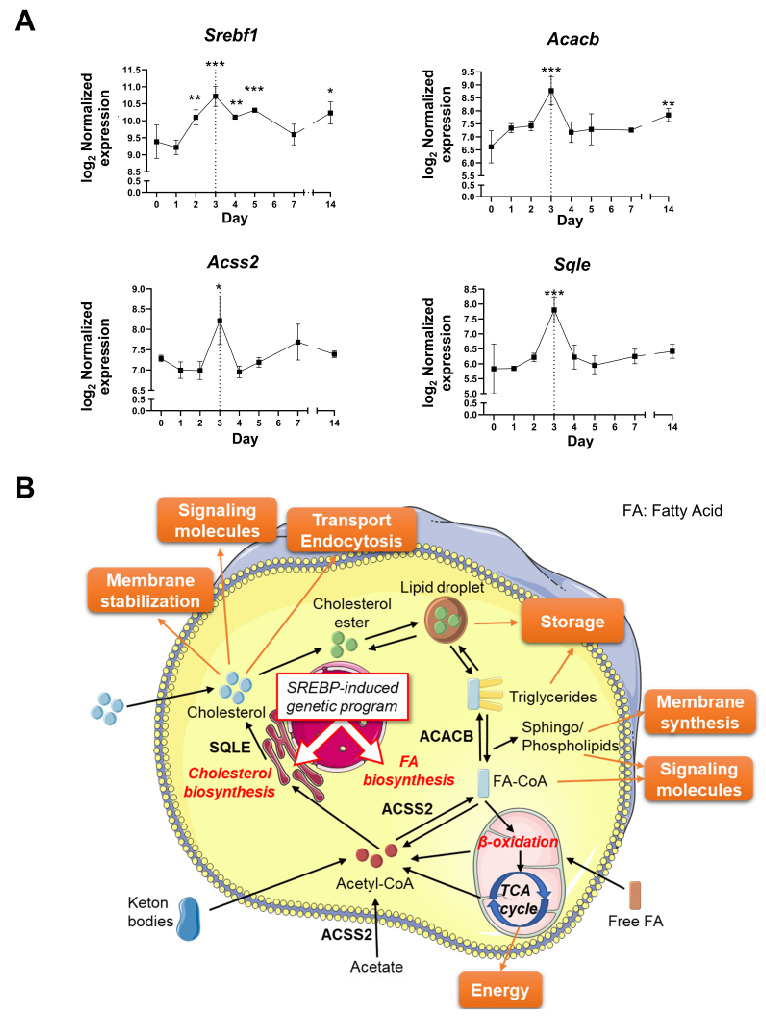
**During the resolution of inflammation, both fatty acid and cholesterol biosynthesis pathways are activated.** (**A**) The expression level of key genes in lipid metabolism reached a peak on day 3 after injury, according to RNA-seq data. Error bars show the mean (SD). Significant differences are represented as * if *p* value *p* < 0.05, ** if *p* < 0.01, and *** if *p* < 0.001. (**B**) In a normal cell, lipid metabolism is regulated by SREBP proteins and encompasses multiple reactions that generate lipids for three main purposes: (1) as source of energy, (2) as membrane constituents, and (3) as signaling/regulatory molecules. (1) Energy is mainly produced by oxidative metabolism through the TCA cycle in the mitochondria. The first step of the TCA cycles uses acetyl-coenzyme A (acetyl-CoA) to generate citrate. Mitochondrial acetyl-CoA can be generated from lipidic sources: by the breakdown of ketone bodies or by the β-oxidation of fatty acyl-CoA (FA-CoA). FA-CoA is obtained by the activation of fatty acids (FAs), which were either synthetized de novo, imported into the cells as free FAs, or obtained by the hydrolysis of triglycerides or cholesterol esters. Triglycerides and cholesterol esters are used for lipid transport or intracellular storage as lipid droplets. (2) Cellular membranes are mainly composed of sphingolipids, phospholipids, and cholesterol, and their respective proportions modulate membrane properties (e.g., lipid rafts). (3) Cholesterol is necessary for the synthesis of steroid hormones and vitamin D, while polyunsaturated fatty acids (PUFAs) are precursors for pro-inflammatory eicosanoids (e.g., leukotrienes, prostaglandins) and specialized pro-resolving mediators (e.g., resolvins, lipoxins). Membrane lipids can also be converted into signaling molecules (e.g., PIP2 → DAG +IP3). After acute injury, upregulation of *Srebf1* encoding SREBP1 promotes the transcription of enzymes involved in the synthesis of acetyl-CoA, fatty acids, and cholesterol including ACSS2, ACACB, and SQLE, respectively. This scheme was partly generated using Servier Medical Art, provided by Servier, licensed under a Creative Commons Attribution 3.0 unported license.

**Figure 7 ijms-24-04309-f007:**
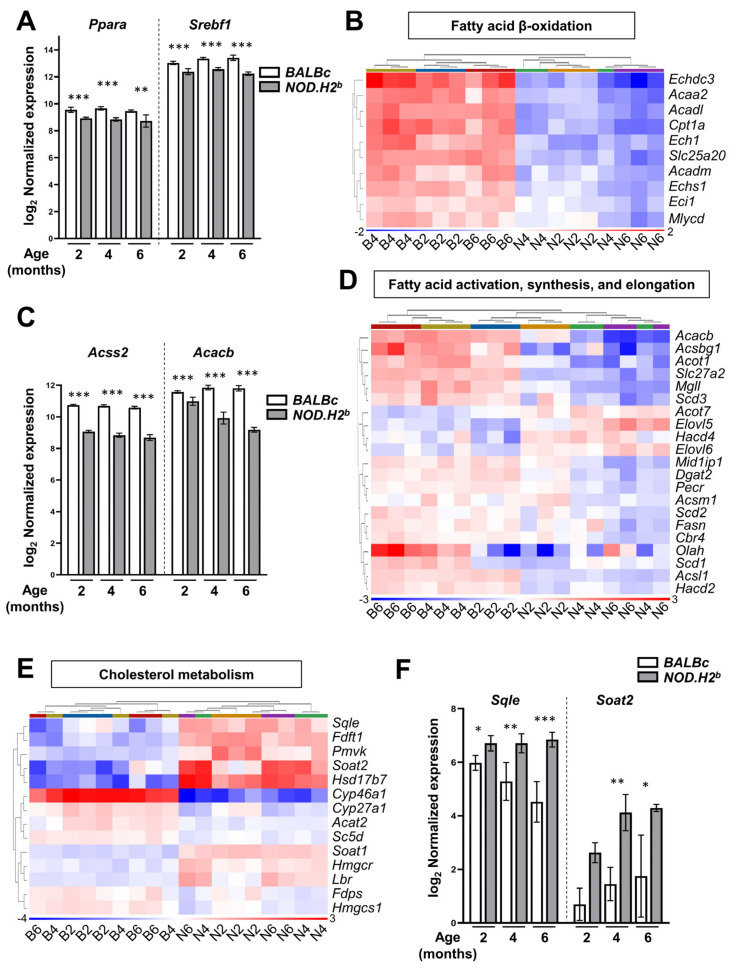
**Chronic inflammation upregulates enzymes related to cholesterol production but downregulates fatty acid metabolism.** RNA-seq data of LGs from *NOD.H2^b^* and *BALBc* mice. Bar plots show the normalized mRNA expression while heatmaps compare expression levels of genes involved in (**A**) the regulation of lipogenic genes, (**B**) mitochondrial fatty acid β-oxidation, (**C**,**D**) fatty acid activation, synthesis, and elongation and (**E**,**F**) cholesterol metabolism. Heatmap sample legends correspond to B = *BALBc* and N= *NOD.H2^b^*, followed by mouse age in months. Error bars show the mean (SD). Significant differences are represented as * if *p* value *p* < 0.05, ** if *p* < 0.01, and *** if *p* < 0.001.

**Figure 8 ijms-24-04309-f008:**
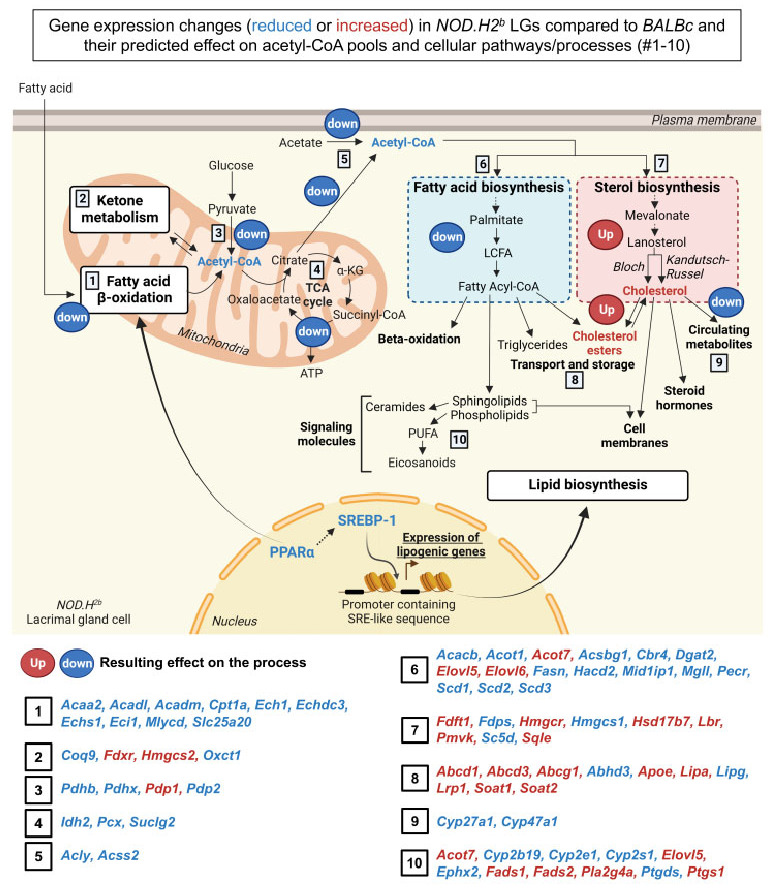
**Lipid metabolism is altered in chronically inflamed LGs.** According to RNA-seq data of LGs from *NOD.H2^b^* and *BALBc* mice, disease progression is associated with an overall downregulation (“down”) of PPARα/SREBP-1 signaling and their downstream target genes involved in fatty acid β-oxidation and TCA (Tricarboxylic acid) cycle happening in the mitochondria, as well as acetyl-CoA/fatty acid biosynthesis that can take place in several cellular compartments. By contrast, genes involved in cholesterol/cholesterol esters biosynthesis are mostly upregulated (“up”) with chronic inflammation. Genes implicated in processes one to ten and significantly altered (upregulated in red, downregulated in blue) during chronic inflammation are shown below. PUFA: Polyunsaturated fatty acid. Created with BioRender.com.

**Table 1 ijms-24-04309-t001:** **Sequences of primers used for qRT-PCR**.

Target	Primer Sequence
*Nlrp3*	Forward: 5′-AGAAGAGACCACGGCAGAAG-3′
	Reverse: 5′-CCTTGGACCAGGTTCAGTGT-3′
*Nlrc4*	Forward: 5′-CTTGGCCAGGAGAGCCTTG-3′
	Reverse: 5′-GGGCTCGTCTGTTGTTCCTT-3′
*Aim2*	Forward: 5′-GATTCAAAGTGCAGGTGCGG-3′
	Reverse: 5′-TCTGAGGCTTAGCTTGAGGAC-3′
*Casp1*	Forward: 5′-GCGAAGCATACTTTCAGTTTC-3′
	Reverse: 5′-TCTCCTTCAGGACCTTGTCG-3′
*Casp4(11)*	Forward: 5′-AGGAGCCCACTCCTACAGAG-3′
	Reverse: 5′-AAGGTTGCCCGATCAATGGT-3′
*Actb*	Forward: 5′-AGAGGGAAATCGTGCGTGAC-3′
	Reverse: 5′-CAATAGTGATGACCTGGCCGT-3′
*Gapdh*	Forward: 5′-CGTCCCGTAGACAAAATG GT-3′
	Reverse: 5′-TTGATGGCAACAATCTCC AC-3′

## Data Availability

All RNAseq data are publicly available from the Gene Expression Omnibus (GEO) database at https://www.ncbi.nlm.nih.gov/geo/ accessed on 19 February 2023 under accession numbers: GSE99093 for acute injury [13], and GSE210332 for chronic inflammation [37].

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
