# Peer review of "Lacrimal Gland Epithelial Cells Shape Immune Responses through the Modulation of Inflammasomes and Lipid Metabolism"

_ijms, 2023, doi:10.3390/ijms24054309_

Round 1
Reviewer 1 Report
Dry eye disease occurs through impaired tear secretion following lacrimal gland (LG) inflammation. As aberrant inflammasome activation occurs in autoimmune disorders including Sjögren’s syndrome (SJ). The authors examined the inflammasome pathway and regulators during acute and chronic LG inflammation. Acute LG injury was induced by interleukin (IL)-1α injection. Chronic inflammation was studied using two SJ models (diseased NOD.H2b vs. healthy BALBc mice, and Thrombospondin-1-null (TSP-1-/-) vs. TSP-1WT C57BL/6J mice). It is shown that acute and chronic inflammation induced inflammasomes in LG epithelial cells and upregulated inflammasome sensors, caspases 1/4, and Il1b and Il18, as well as increased IL-1β maturation in SJ models compared to control LG. Using RNA-seq data of regenerating lacrimal glands, they found that lipogenic genes were upregulated during the resolution of inflammation following acute injury. In chronically inflamed NOD.H2b LG, altered lipid metabolism was associated with disease progression. Genes involved in mitochondrial metabolism and fatty acid synthesis including Peroxisome Proliferator Activated Receptor Alpha (PPARα)/Sterol Regulatory Element Binding 1 (SREBP-1)-dependent signaling were downregulated, but cholesterol metabolism was increased in NOD.H2b lacrimal glands. The authors conclude that epithelial cells can promote immune responses by forming inflammasomes. Sustained inflammasome activation with altered lipid metabolism are key players of SJ-like pathogenesis in the NOD.H2b mouse LG by promoting epithelial dysfunction and inflammation. The paper contains new important mechanistic information, is well thought out and written, and the data are credible and significant.
This reviewer has only some minor concerns about the manuscript.
1. Please add to the abstract the induction of inflammasome formation by lipopolysaccharide and nigericin.
2. In the immunostaining section (Methods), please indicate which protein was used as an inflammasome marker.
3. Please change p-adj≤0.05 to p-adj<0.05 (line 216).
4. The authors might like to present Fig. 1B and C at higher magnification to better appreciate the expression of ASC.
5. It is customary to add asterisk explanations as p values in the figure legends.
6. In Fig. 2H, please add Greek beta before actin.
7. Please abbreviate qRT-PCR, not RT-qPCR.
8. In Fig. 4B please state that data from individual mice are shown.
9. Please explain whether the color of nodes in Fig. 5C is related to up or down regulation.
Author Response
Referee 1.
Dry eye disease occurs through impaired tear secretion following lacrimal gland (LG) inflammation. As aberrant inflammasome activation occurs in autoimmune disorders including Sjögren’s syndrome (SJ). The authors examined the inflammasome pathway and regulators during acute and chronic LG inflammation. Acute LG injury was induced by interleukin (IL)-1α injection. Chronic inflammation was studied using two SJ models (diseased NOD.H2b vs. healthy BALBc mice, and Thrombospondin-1-null (TSP-1-/-) vs. TSP-1WT C57BL/6J mice). It is shown that acute and chronic inflammation induced inflammasomes in LG epithelial cells and upregulated inflammasome sensors, caspases 1/4, and Il1b and Il18, as well as increased IL-1β maturation in SJ models compared to control LG. Using RNA-seq data of regenerating lacrimal glands, they found that lipogenic genes were upregulated during the resolution of inflammation following acute injury. In chronically inflamed NOD.H2b LG, altered lipid metabolism was associated with disease progression. Genes involved in mitochondrial metabolism and fatty acid synthesis including Peroxisome Proliferator Activated Receptor Alpha (PPARα)/Sterol Regulatory Element Binding 1 (SREBP-1)-dependent signaling were downregulated, but cholesterol metabolism was increased in NOD.H2b lacrimal glands. The authors conclude that epithelial cells can promote immune responses by forming inflammasomes. Sustained inflammasome activation with altered lipid metabolism are key players of SJ-like pathogenesis in the NOD.H2b mouse LG by promoting epithelial dysfunction and inflammation. The paper contains new important mechanistic information, is well thought out and written, and the data are credible and significant.
This reviewer has only some minor concerns about the manuscript.
- Please add to the abstract the induction of inflammasome formation by lipopolysaccharide and nigericin.
We thank the reviewer for this suggestion and added the LPS/nigericin experiment to the abstract (l19-20, l25). We also provided slightly more details about the technical approaches used to study inflammasome activation (l23-25).
- In the immunostaining section (Methods), please indicate which protein was used as an inflammasome marker.
In accordance with the reviewer’s comment, we specified in the methods the detection of the ASC-Citrine fusion protein for the detection of inflammasome complexes (l175-178). To improve clarity, we mentioned the targets of each antibody used in this study in the material and methods (169-173) and in the main text (l.273-276).
Materials and methods:
The following primary antibodies were used for immunostaining overnight at 4°C: mouse monoclonal α-smooth muscle actin antibody (1/200, clone 1A4; #A2547, RRID:AB_476701, Millipore-Sigma) was used to label MECs and mural cells, rat monoclonal CD31 antibody (1/100, #553370, RRID:AB_394816, BD Biosciences) was specific to blood vessels, mouse monoclonal E-Cadherin antibody (1/200, #610182, RRID:AB_397581, BD Biosciences) labeled epithelial cells, and rabbit polyclonal AIM2 antibody (1/100, #63660, RRID:AB_2890193, Cell Signaling Technology) was used to detect AIM2-inflammasomes.
Appropriate fluorochrome-conjugated secondary antibodies were obtained from Invitrogen (Waltham, MA, USA) and nuclei were counterstained with DAPI. The formation of inflammasome complexes following bacterial and sterile stimuli in WT mice and during chronic inflammation in TSP-1-/- mice was detected using the ASC-Citrine fusion protein that is constitutively and ubiquitously expressed in R26ASC-citrine mice and forms fluorescent specks upon inflammation.
Results:
Anti-α-SMA immunostaining was used to identify myoepithelial cells (MECs) that surround the secretory units formed by acinar cells and pericytes around blood vessels (blood vessels were labeled by antibody to CD31) (Fig. 1B). Therefore, tubular structures negative for CD31 and α-SMA staining were identified as ducts.
- Please change p-adj≤0.05 to p-adj<0.05 (line 216).
Thank you for pointing out this mistake. We corrected the text accordingly (l. 234, 500).
- The authors might like to present Fig. 1B and C at higher magnification to better appreciate the expression of ASC.
Following the reviewer’s suggestion, we provided in Fig. S1A, C panels with enlarged micrographs of relevant areas.
- It is customary to add asterisk explanations as p values in the figure legends.
We added “Significant differences are represented as * if p value p < 0.05, ** if p < 0.01 and *** if p < 0.001.” in the legend of relevant figures.
- In Fig. 2H, please add Greek beta before actin.
Thank you for noticing this, we corrected this figure accordingly.
- Please abbreviate qRT-PCR, not RT-qPCR.
The abbreviation has been modified in the manuscript according to the reviewer’s comment.
- In Fig. 4B please state that data from individual mice are shown.
To improve clarity, we modified the legend as following (l482-483):
Expression heatmap of the genes involved in “interleukin-1 production” (GO:0032612) pathway in each biological replicate (1 sample = 1 LG) from BALBc (“B”) or NOD.H2b (“N”) mice at 2, 4 or 6M. Individual LG samples cluster in 2 main groups: all NOD.H2b samples show an important activation of genes involved in this pathway compared to BALBc.
Similarly, we improved the legend for Fig.3 (l.468):
Expression heatmap of the genes involved in the pathway entitled “interleukin-1 production” (GO:0032612) shows that individual LG samples cluster in 3 main groups, with biological replicates from days 1 and 2 after injury presenting an important activation of a set of genes involved in this pathway.
- Please explain whether the color of nodes in Fig. 5C is related to up or down regulation.
The ontology terms were obtained by analyzing the list of 66 upregulated genes. Each term is represented by a circle node, where its size is proportional to the number of input genes that fall under that term, and its color represents its cluster identity.
We agree with the reviewer’s comment that the legend lacked details about how the network was built, and modified it as following (l.565-569):
One term from each cluster is selected to have its term description shown as cluster identity. Nodes that share the same cluster identity are typically close to each other and have the same color (i.e., nodes of the same color belong to the same cluster). The node size reflects the number of input genes that fall under that term. The thickness of the edge represents the similarity score (> 0.3).
Reviewer 2 Report
The authors of manuscript ID: ijms-2156699 titled „ Lacrimal gland epithelial cells shape immune responses through the modulation of inflammasomes and lipid metabolism” presented interesting and very valuable research results.The lacrimal gland plays a pivotal role in keeping the ocular surface lubricated, and protecting it from environmental exposure and insult. Current therapies do not treat the underlying deficiency of the lacrimal gland, but merely provide symptomatic relief. To develop more sustainable and physiological therapies, such as in vivo lacrimal gland regeneration or bioengineered lacrimal gland implants, a thorough understanding of lacrimal gland development at the molecular level is of paramount importance. Based on the structural and functional similarities between rodent and human eye development, extensive studies have been undertaken to investigate the signaling and transcriptional mechanisms of lacrimal gland development using mouse as a model system. As aberrant inflammasome activation occurs in autoimmune disorders including Sjögren’s syndrome, authors analyzed the inflammasome pathway during acute and chronic inflammation and investigated its potential regulators. Acute injury of the lacrimal gland was induced by interleukin (IL)-1α injection and chronic inflammation was studied using two Sjögren’s syndrome models (diseased NOD.H2b compared to healthy BALBc mice, and Thrombospondin-1-null (TSP-1- /-) compared to TSP-1WT C57BL/6J mice). The authors have shown that the lacrimal gland epithelium plays a pivotal role in the innate immune response by activating inflammasome signaling in response to ex- ogenous or endogenous stimuli. Dysregulation of this protective defense mechanism during chronic inflammation is associated with an imbalance between the metabolic pathways producing fatty acids and cholesterol.
Overall this is a well-written manuscripts and would add significant new knowledge to the field, with implications for drug discovery for ocular surface disease.Congratulations and thanks to the authors for preparing a very good manuscript
Author Response
Referee 2
The authors of manuscript ID: ijms-2156699 titled „ Lacrimal gland epithelial cells shape immune responses through the modulation of inflammasomes and lipid metabolism” presented interesting and very valuable research results.
The lacrimal gland plays a pivotal role in keeping the ocular surface lubricated, and protecting it from environmental exposure and insult. Current therapies do not treat the underlying deficiency of the lacrimal gland, but merely provide symptomatic relief. To develop more sustainable and physiological therapies, such as in vivo lacrimal gland regeneration or bioengineered lacrimal gland implants, a thorough understanding of lacrimal gland development at the molecular level is of paramount importance. Based on the structural and functional similarities between rodent and human eye development, extensive studies have been undertaken to investigate the signaling and transcriptional mechanisms of lacrimal gland development using mouse as a model system. As aberrant inflammasome activation occurs in autoimmune disorders including Sjögren’s syndrome, authors analyzed the inflammasome pathway during acute and chronic inflammation and investigated its potential regulators. Acute injury of the lacrimal gland was induced by interleukin (IL)-1α injection and chronic inflammation was studied using two Sjögren’s syndrome models (diseased NOD.H2b compared to healthy BALBc mice, and Thrombospondin-1-null (TSP-1- /-) compared to TSP-1WT C57BL/6J mice). The authors have shown that the lacrimal gland epithelium plays a pivotal role in the innate immune response by activating inflammasome signaling in response to ex- ogenous or endogenous stimuli. Dysregulation of this protective defense mechanism during chronic inflammation is associated with an imbalance between the metabolic pathways producing fatty acids and cholesterol.
Overall this is a well-written manuscripts and would add significant new knowledge to the field, with implications for drug discovery for ocular surface disease.
Congratulations and thanks to the authors for preparing a very good manuscript.
We sincerely thank the reviewer for this very positive feedback.
Reviewer 3 Report
This manuscript reports the activation of inflammasomes and the alteration of lipid metabolism in lacrimal glands during acute and chronic inflammation. These findings are significant to the field and most data are solid. Following issues need be addressed before the consideration of publication.
Major point:
1. In Figure 1B-K, many ASC-cit+ cells appear in the stromal and most ASC-cit+ cells are a-Sma-. Co-staining of an epithelial marker seems necessary to support the statement “Epithelial cells can sense microbial/sterile inflammatory stimuli”.
2. NOD.H2b mice at 2 months old (2M) are generally considered at the pre-disease stage without obvious leukocyte infiltration in lacrimal glands (Coursey et al., 2017). Therefore, in Figures 4 and 7, it seems necessary to also compare 2M data with 4M and 6M data.
Minor point:
1. Data not shown should be added as supplementary data.
2. In Figure 1B-C, images with higher magnifications are needed and ASC-cit+ cells need be quantified.
3. “log2(FC) = ± log2(1.5)” should be “log2(FC) = ± 1.5”
4. Please specify how the normalized mRNA expression was calculated using RNA-seq data in Figures 3-7.
Author Response
Referee 3
This manuscript reports the activation of inflammasomes and the alteration of lipid metabolism in lacrimal glands during acute and chronic inflammation. These findings are significant to the field and most data are solid. Following issues need be addressed before the consideration of publication.
Major point:
- In Figure 1B-K, many ASC-cit+cells appear in the stromal and most ASC-cit+ cells are a-Sma-. Co-staining of an epithelial marker seems necessary to support the statement “Epithelial cells can sense microbial/sterile inflammatory stimuli”.
ASC-specks were indeed formed in some stromal cells, and the majority of specks in the epithelium were found in SMA- acinar cells. To improve clarity about our staining strategy, we added some explanations as detailed in the answer to point 2 from reviewer 1. We also provided enlarged micrographs in Fig. S1A-D to better illustrate our observations. We hope these changes better support our conclusions.
- NOD.H2b mice at 2 months old (2M) are generally considered at the pre-disease stage without obvious leukocyte infiltration in lacrimal glands (Coursey et al., 2017). Therefore, in Figures 4 and 7, it seems necessary to also compare 2M data with 4M and 6M data.
We recently published a study about disease progression in NOD.H2b mice using different approaches (flow cytometry, IHC, bulk RNAseq, spatial transcriptomics) including the RNAseq data analyzed in this manuscript (Mauduit et al, 2022). In agreement with the reviewer’s comment, we showed that clinical disease is not fully and consistently established in 2M NOD.H2b males. However, they already have many alterations in their LG transcriptome and the number of B/T lymphocytes in the LG is significantly increased compared to healthy controls (similar to the results of Coursey et al., 2017 and our previous publication Mauduit et al, 2022). To clarify the biological context for the RNAseq data, we modified the paragraph at l.403-410:
We also analyzed transcriptomic changes leading to inflammasome priming and activation during the development of chronic inflammation. To do so, we mined our previously published RNAseq data of LGs from 2M, 4M, and 6M NOD.H2b (diseased) and BALBc (control) males (GSE210332) (37). In this study, we showed that although there is a major shift in gene expression between 2M and 4M/6M mice, 2M NOD.H2b LGs already feature many alterations at the transcriptomic level compared to BALBc controls. We also found that B and T cell infiltrates appear as early as 2M (early stage of the disease) in NOD.H2b males - although not to the same extent as 6M males (clinical stage).
While we included data from 2M, 4M and 6M mice in all plots, we did not add a direct comparison of NOD mice on the main figures because we think that proper comparisons should systematically be done using age-matched BALBc controls to circumvent potential inherent age-induced changes. However, we agree with the reviewer that this aspect should be added to the manuscript.
Therefore, we added to Fig. S3 the expression heatmap showing genes from “interleukin-1 production” (GO:0032612) pathway that are significantly changed between 2M and 4M NOD.H2b LGs. We also included all the fold-changes and p-adj obtained for every comparison (2M/4M, 2M/6M, 4M/6M) into Table S2. Following the reviewer’s recommendation, we added the interpretation below:
While most of inflammasome-related genes were significantly upregulated at 2M compared to healthy controls, the expression level of some of them (including Aim2, Nlrc4, Nlrp1a, Casp1, Panx1 and Gsdmd) increased further at 4M (Fig. S3C). By contrast, none of the genes from the “Interleukin-1 production” pathway passed our thresholds when comparing 4M and 6M NOD.H2b LGs. Moreover, no DEGs were found for the 2M/4M and 4M/6M comparisons in BALBc LGs. This shows that, in NOD.H2b LGs, activation of the inflammasome pathway amplifies up to 4M, age at which the mechanisms of chronic inflammation are established (37).
For the part about lipid metabolism during chronic inflammation (Fig. 7), we added at l.631-633 this comment:
The number of significant DEGs in NOD.H2b LGs and the extent of the alterations in their expression level compared to their respective BALBc controls increased with age, especially between 2M and 4M (Fig. 7C-D, Table S4).
We also added a second sheet into Table S4 to show the fold changes and p-values of all genes for every comparison between NOD.H2b groups. The main text was modified according to this new data at l.662-665:
Few of these alterations significantly aggravated between 2M and 4M/6M NOD.H2b mice (Table S4). Only Acacb was further downregulated at each time-point, while not being affected in BALBc mice. Altogether, this suggests that the changes in lipid metabolism are early mechanisms of disease development.
We thank the reviewer for his/her valuable suggestion and hope these modifications adequately addressed his/her concerns.
Reference:
Mauduit O, Delcroix V, Umazume T, de Paiva CS, Dartt DA, Makarenkova HP. Spatial transcriptomics of the lacrimal gland features macrophage activity and epithelium metabolism as key alterations during chronic inflammation. Front Immunol. 2022 Oct 17;13:1011125. doi: 10.3389/fimmu.2022.1011125. PMID: 36341342; PMCID: PMC9628215.
Minor point:
- Data not shown should be added as supplementary data.
We believe this data do not really add much information to this study, and since we do not compare the different ages at this stage of the manuscript, we decided to minimize the number of supplemental figures and removed this sentence.
- In Figure 1B-C, images with higher magnifications are needed and ASC-cit+cells need be quantified.
As mentioned in point 1, we added higher magnifications of relevant areas from Fig.1 as supplemental data Fig. S1. Because the experiment with LPS+nigericin was rather a verification of the model and since the main result was the capacity of epithelial cells to form inflammasomes, we added the quantification of ASC specks to supplementary Fig. S1C.
- “log2(FC) = ± log2(1.5)” should be “log2(FC) = ± 1.5”
We used a FC=1.5 cut-off equal to log2(FC) = ± log2(1.5) ≈ ± 0.585. To avoid any confusion, we added this information in the manuscript at l. 234:
For all projects, differentially expressed genes (DEGs) were selected based on a fold-change (FC) cut-off equal to log2(FC) ≈ ± 0.585 (corresponding to FC = 1.5) and p-adj < 0.05.
- Please specify how the normalized mRNA expression was calculated using RNA-seq data in Figures 3-7.
This information was added in the material and methods section at l.231-232:
ROSALIND uses DEseq2 R library to normalize read counts and calculate fold-changes along with corresponding p-values, as previously described [37].
Round 2
Reviewer 3 Report
Most concerns were satisfactorily addressed. However, no supplementary figures were found in the supplementary files. Therefore, I can't evaluate revisions related to supplementary figures now.